



# ChAP 1.0: A stationary tropospheric sulphur cycle for Earth system models of intermediate complexity

Alexey V. Eliseev[1,2,3,4], Rustam D. Gizatullin[4], and Alexandr V. Timazhev[2]

[1]Lomonosov Moscow State University, Faculty of Physics, Moscow, Russia
[2]A.M. Obukhov Institute of Atmospheric Physics, Russian Academy of Sciences, Moscow, Russia
[3]Institute of Applied Physics, Russian Academy of Sciences, Nizhny Novgorod, Russia
[4]Kazan Federal University, Kazan, Russia

**Correspondence:** A.V Eliseev (eliseev.alexey.v@gmail.com)

**Abstract.** A stationary, computationally efficient scheme ChAP-1.0 (Chemical and Aerosol Processes, version 1.0) for the sulphur cycle in the troposphere is developed. This scheme is designed for Earth system models of intermediate complexity (EMICs). The scheme accounts for sulphur dioxide emissions into the atmosphere, its deposition to the surface, oxidation to sulphates, and dry and wet deposition of sulphates on the surface. The calculations with the scheme are performed forced

by anthropogenic emissions of sulphur dioxide into the atmosphere for 1850-2000 adopted from the CMIP5 dataset and by the ERA-Interim meteorology assuming that natural sources of sulphur into the atmosphere remain unchanged during this period. The ChAP output is compared to changes of the tropospheric sulphur cycle simulations: with the CMIP5 data, with the IPCC TAR ensemble, and with the ACCMIP phase II simulations. In addition, in regions of strong anthropogenic sulphur pollution, ChAP results are compared to other data, such as the CAMS reanalysis, EMEP MSC-W, and with individual model

simulations. Our model reasonably reproduces characteristics of the tropospheric sulphur cycle known from these information sources. In our scheme, about half of the emitted sulphur dioxide is deposited to the surface and the rest in oxidised into sulphates. In turn, sulphates are mostly removed from the atmosphere by wet deposition. The lifetime of the sulphur dioxide and sulphates in the atmosphere is close to 1 day and 5 days, respectively. The limitation of the scheme are acknowledged and the prospects for future development are figured out. Despite its simplicity, ChAP may be successfully used to simulate

anthropogenic sulphur pollution in the atmosphere at coarse spatial and time scales.

## 1 Introduction

Sulphur compounds in the troposphere are important pollutants and contribute both to the direct radiative effect, which is also known as an aerosol-radiation interaction, and, owing to their hygroscopicity, are major contributors to the aerosol indirect effects on climate – the so called aerosol-cloud interaction (Charlson et al., 1992; Boucher et al., 2013). The direct radiative

forcing from the preindustrial till 2010's is estimated to amount from $-0.2$ to $-0.8 \, \mathrm{W\,m^{-2}}$ (Boucher et al., 2013; Myhre et al., 2013; Shindell et al., 2013; Zelinka et al., 2014; Matus et al., 2019). The sulphate contribution to aerosol-cloud interaction leads to the corresponding indirect forcing from -0.2 to $-1.2 \, \mathrm{W\,m^{-2}}$ (Zelinka et al., 2014; McCoy et al., 2017). These forcings arise from the anthropogenic release of sulphates and of sulphur precursors (mostly of sulphur dioxide $SO_2$).





Apart influencing climate changes, sulphur compounds impact terrestrial vegetation and, thus, the global carbon cycle.
The first effect is due to suppression of the terrestrial vegetation gross primary production arising from uptake of sulphur
dioxide by leaves with subsequent injury of photosynthesis tissues of plants (Semenov et al., 1998). This suppression may be
as large as 10 per cent relative to the $SO_2$-unaffected plants, especially in moist tropical forests (Eliseev, 2015a, b; Eliseev et al.,
2019). Another impact is due to acidification of soils and surface waters with a risk of vegetation poisoning (Kuylenstierna
et al., 2001).

There is a number chemically active sulphur species in the Earth atmosphere. Among those, the most abundant are sulphur
dioxide $SO_2$, which is either oxidised from precursors such as dimethyl sulphide (DMS), hydrogen sulphide $H_2S$, and carbon
disulphide $CS_2$, or emitted by volcanos, or released due to anthropogenic activity (Seinfeld and Pandis, 2006; Warneck, 2000;
Surkova, 2002). Additional minor $SO_2$ source is due to biomass burning. Less abundant, but still important in global sulphur
cycle are dimethyl sulphide, dimethyl sulfoxide (DMSO), and methanesulphonic acid (MSA). All these species chemically
interact with each other and undergo wet and dry deposition on the Earth surface. As a whole, chemical reaction chain converts
sulphur compounds into sulphur dioxide, which is further oxidised into sulphuric acid $H_2SO_4$ and sulphates $SO_4^{2-}$ (Seinfeld
and Pandis, 2006; Warneck, 2000; Surkova, 2002).

All this motivated researchers to implement interactive sulphur cycle into global climate models (or, more precisely, into
Earth system models, ESMs). Starting from the pioneering paper by Chatfield and Crutzen (1984), research groups from
different modelling centres included sulphur cycles into their ESMs. The most active phase of these projects was in late
1980's and 1990's, which resulted in a number of chemical-transport models which may be or may be not coupled to ESMs:
MOGUNTIA (Langner and Rodhe, 1991), IMAGES (Pham et al., 1995), ECHAM (Feichter et al., 1996; Roelofs et al., 1998),
Harvard-GISS (Koch et al., 1999), CCM1-GRANTOUR (Chuang et al., 1997), CCM3 (Barth et al., 2000; Rasch et al., 2000),
CCCMA (Lohmann et al., 1999), and GOCART (Chin et al., 2000). These models were summarised in the Intergovernmental
Panel on Climate Change Third Assessment Report (IPCC TAR) (Houghton et al., 2001, their Table 5.8). Later, these models
became able to account for other types of aerosol and interaction between different geochemical cycles (Forster et al., 2007;
Boucher et al., 2013), which led to development of the AeroCom and ACCMIP (Atmospheric Chemistry and Climate Model
Intercomparison Project) activities, (Shindell et al., 2013; Lamarque et al., 2013a; Myhre et al., 2013; Tsigaridis et al., 2014;
Fiedler et al., 2019; Riemer et al., 2019; Bellouin et al., 2020; Gliß et al., 2021).

In parallel, a number of the reduced-complexity, computationally cheap ESMs, which are collectively referred to as Earth
system models of intermediate complexity (EMICs), has emerged (Claussen et al., 2002; Petoukhov et al., 2005; Eby et al.,
2013; Zickfeld et al., 2013; MacDougall et al., 2020). These models are mostly targeted for simulations at coarse (e.g., sub-
continental) spatial scales but are ran either for very long time intervals or for large ensemble simulations (e.g., Eby et al.,
2009; Collins et al., 2011; Eliseev, 2011; Willeit et al., 2014; MacDougall and Knutti, 2016; Ganopolski and Brovkin, 2017;
Muryshev et al., 2017). One may argue that such models also needs modules to mimic the atmospheric chemistry. For instance,
lacking interactive atmospheric sulphur cycle, EMICs attempt to simulate the 20th century climate changes ignoring radiative
forcing from tropospheric sulphates. This hampers an evaluation of the realism of the models of this type. At the date, there are
only two EMICs, which implemented radiative forcing from sulphates: IAPRAS-MSU (A.M. Obukhov Institute of the Atmo-





spheric Physics, Russian Academy of Sciences – Lomonosov Moscow State University) (Eliseev et al., 2007) and Climber-2
(Bauer et al., 2008).

The latter model also implements a very simple atmospheric sulphur cycle scheme, in which sulphate burden per unit area
is related to their precursor emissions at the same grid cell by using a prescribed coefficient, which, in turn, is related to
atmospheric lifetimes of sulphates and their precursors taking into account that part of the emitted precursors are deposited
before they are oxidised into sulphates. No horizontal transport of sulphates and their chemical precursors are allowed for. This
approach is reasonable for Climber-2 with its very coarse horizontal resolution ($10°$ by latitude and $51.3°$ on longitude, Bauer
et al., 2008), but becomes problematic for other EMICs, in which this resolution is higher.

Somewhat similar, but inverse approach was pursued in the IAPRAS-MSU model. In this model, $SO_4$ burden is prescribed
as a function of time, and $SO_2$ burden is reconstructed by using an atmospheric moisture-dependent coefficient to calculate the
$SO_2$ impact on terrestrial gross primary production (Eliseev et al., 2019).

The goal of the present paper is to make a step beyond the Climber-2 and IAPRAS-MSU approaches and to allow for
transport of sulphur species in the horizontal direction and to calculate characteristics of the sulphur cycle directly. This should
be done in a computationally efficient manner in order not to destroy an important property of EMICs – their small turnaround
time. This precludes usage of the sulphur cycle scheme implemented into the above-mentioned chemical transport model.
Thus, we developed a stationary scheme, ChAP (Chemistry and Aerosol Processes), which is able to mimic gross dynamics
of the atmospheric chemistry. Its contemporary version, ChAP-1.0 implements only the anthropogenic part of the atmospheric
sulphur cycle, but we plan to extend the scheme in future.

Below, a theoretical background for our scheme is presented and its offline performance is tested.

## 2 Scheme description

### 2.1 General considerations

We start from the general equations governing mass concentrations of $SO_2$, $q_{SO_2}$, and $SO_4$, $q_{SO_4}$ (Seinfeld and Pandis, 2006;
Warneck, 2000; Surkova, 2002):

$$
\begin{aligned}
\frac{\partial q_{SO_2}}{\partial t} + \boldsymbol{U} \cdot \nabla q_{SO_2} &= e_{SO_2} + r_{SO_2,prod} - r_{in-cl} - r_{gas} - d_{SO_2,dry} - d_{SO_2,wet} + a_{SO_2}, \\
\frac{\partial q_{SO_4}}{\partial t} + \boldsymbol{U} \cdot \nabla q_{SO_4} &= e_{SO_4} + r_{in-cl} + r_{gas} - d_{SO_4,dry} - d_{SO_4,wet} + a_{SO_4},
\end{aligned}
\tag{1}
$$

where $e_Y$ is emission rate for substance $Y$ ($Y \in \{SO_2, SO_4\}$), $\boldsymbol{U}$ is three–dimensional transport velocity, $r_{in-cl}$ and $r_{gas}$
are, respectively, in-cloud and gas–phase oxidation converting $SO_2$ to $SO_4$, $r_{SO_2,prod}$ is chemical $SO_2$ production rate in the
atmosphere, $d_{Y,dry}$ and $d_{Y,wet}$ are dry and wet deposition rates for substance $Y$, correspondingly, and $a_Y$ are diffusive and
convective redistributions of $Y$.

Because our goal is to develop a scheme for sufficiently large time steps, we assume that vertical profiles of both substances
are universal in a sense that $q_Y$ at each altitude (as well as the total burden of $Y$) depends only on the respective surface
value. We assume an exponential dependence of $q_Y$ on geometrical altitude (thus, explicitly excluding stratospheric sulphur





compounds) with the vertical scales $H_Y$ which is of the order of $1\,\mathrm{km}$ (Jaenicke, 1993; Warneck, 2000). The latter leads to the relation between the near–surface mass concentrations $q_{Y,\mathrm{s}}$ and the total burden $B_Y$ of substance $Y$ per unit area:

$$B_Y = q_{Y,\mathrm{s}} H_Y. \tag{2}$$

This allows us to integrate (1) over vertical coordinate, formally from the surface up to the infinity. To simplify the setup, we neglect the dependence of horizontal velocity on the vertical coordinate. The resulting equations read


$$\frac{\partial B_{\mathrm{SO}_2}}{\partial t} + \boldsymbol{U} \cdot \nabla B_{\mathrm{SO}_2} = E_{\mathrm{SO}_2} + R_{\mathrm{SO}_2,\mathrm{prod}} - R_{\mathrm{in-cl}} - R_{\mathrm{gas}} - D_{\mathrm{SO}_2,\mathrm{dry}} - D_{\mathrm{SO}_2,\mathrm{wet}} + A_{\mathrm{SO}_2},$$
$$\frac{\partial B_{\mathrm{SO}_4}}{\partial t} + \boldsymbol{U} \cdot \nabla B_{\mathrm{SO}_4} = E_{\mathrm{SO}_4} + R_{\mathrm{in-cl}} + R_{\mathrm{gas}} - D_{\mathrm{SO}_4,\mathrm{dry}} - D_{\mathrm{SO}_4,\mathrm{wet}} + A_{\mathrm{SO}_4}, \tag{3}$$

where $Y \in \{\mathrm{SO}_2,\ \mathrm{SO}_4\}$,

$$E_Y = \int_0^\infty e_Y \, dz,$$

$$R_{Y,Z} = \int_0^\infty r_{Y,Z} \, dz; \qquad Z \in \{\mathrm{prod,\ in-cl,\ gas}\},$$

$$D_{Y,Z} = \int_0^\infty d_{Y,Z} \, dz; \qquad Z \in \{\mathrm{wet,\ dry}\},$$

$$A_{Y,Z} = \int_0^\infty a_Y \, dz;$$

Eq. (3) is similar to (1) with two important differences: now $U$ is two–dimensional (we choose it at a representative altitude), and $A_Y$ represents only horizontal diffusion.

Further, we assume that major chemical reactions follow the common first–order kinetics relative to the source compounds. In a similar fashion, we assume that sink terms are proportional to the respective burdens. All this leads to

$$R_{\mathrm{in-cl}} = k_{\mathrm{in-cl}} B_{\mathrm{SO}_2},$$
$$R_{\mathrm{gas}} = k_{\mathrm{gas}} B_{\mathrm{SO}_2}, \tag{4}$$
$$D_{Y,\mathrm{Z}} = k_{Y,\mathrm{Z}} B_Y, \qquad Y \in \{\mathrm{SO}_2,\ \mathrm{SO}_4\}; \qquad Z \in \{\mathrm{wet,\ dry}\}).$$

Here $k$'s stand either for the respective reaction rate constants or for the loss rate coefficients.

Based on (Warneck, 2000) and on individual model simulations summarised in (Houghton et al., 2001, their Table 5.5), we
neglect the following terms in (1)

– both non–stationary terms $\partial B_Y / \partial t$,

– chemical $\mathrm{SO}_2$ production in the atmosphere: $R_{\mathrm{SO}_2,\mathrm{prod}} = 0$ (this assumption basically removes part of the natural sources of sulphur dioxide, e.g., the DMS oxidation),

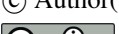



  – gas–phase sulphur dioxide oxidation: $R_{\mathrm{gas}} = 0$,

– wet deposition of sulphur dioxide: $D_{\mathrm{SO_2,wet}} = 0$ (or, equivalently, $k_{\mathrm{SO_2,wet}} = 0$),

This and the previous assumption sets make Eqs. (3) linear with respect to prognostic variables. For time being, we additionally drop diffusion terms $A_Y$. Thus, Eqs. (3) are reduced to

$$\boldsymbol{U} \cdot \nabla B_{\mathrm{SO_2}} = E_{\mathrm{SO_2}} - k_{\mathrm{SO_2}} B_{\mathrm{SO_2}},$$
$$\boldsymbol{U} \cdot \nabla B_{\mathrm{SO_4}} = k_{\mathrm{in-cl}} B_{\mathrm{SO_2}} - k_{\mathrm{SO_4}} B_{\mathrm{SO_4}}, \tag{5}$$

Here, $k_{\mathrm{SO_2}} = k_{\mathrm{in-cl}} + k_{\mathrm{SO_2,dry}}$, $k_{\mathrm{SO_4}} = k_{\mathrm{SO_4,dry}} + k_{\mathrm{SO_4,wet}}$, and $E_{\mathrm{SO_2}}$ is $\mathrm{SO_2}$ emission rate per unit area.

To get a guide, consider an one–dimensional problem with $U = |u| = const$ and the emission localised in the interval $0 \le x \le L$, where $x$ is coordinate in the direction of $U$, and $L$ is the horizontal source size (Jacob, 2000). Its solution are shown in supplementary Fig. S1. The averaged over the $0 \le x \le L$ domain the solution of the first equation (5) reads

$$\overline{B_Y} = \frac{E_Y}{k_Y} \left[ 1 - \frac{1}{\gamma_Y} \left( 1 - e^{-\gamma_Y} \right) \right], \tag{6}$$

where $\gamma_Y = k_Y L / U$. Downwind of the emission region this solution reads

$$B_Y(x) = B_Y(L) \left[ 1 - \exp \left( \gamma_Y - k_Y x / U \right) \right], \tag{7}$$

Eq. (7) allows, in particular, to estimate the horizontal scale of influence for this grid cell source as

$$L_Y \sim 2U / k_Y = 2U \, \mathcal{T}_Y, \tag{8}$$

where $\mathcal{T}_Y = k_Y^{-1}$ is the lifetime of species $Y$ in the atmosphere.

An estimate of $\mathcal{T}_{\mathrm{SO_2}}$ may be obtained from Eq. (8) by using the ERA–Interim data (Dee et al., 2011). For this dataset, the
typical values of zonal $u$ and meridional $v$ velocities in the lower troposphere in the middle latitudes, where the $\mathrm{SO_2}$ pollution is most marked, are up to $7\,\mathrm{m\,s^{-1}}$ and up to $5\,\mathrm{m\,s^{-1}}$ (Fig. S2). Because the typical $\mathrm{SO_2}$ lifetime is around 1-2 days (Warneck, 2000; Surkova, 2002; Houghton et al., 2001, their Table 5.5), we can estimate $L_{\mathrm{SO_2}} \sim 10^3$ km (somewhat larger in the zonal direction and somewhat smaller in the meridional one). This corresponds to few grid cells provided that the grid cell size is several hundred kilometres.

Typical horizontal scale for $\mathrm{SO_4}$, $L_{\mathrm{SO_4}}$, may be estimated in a similar way. Assuming that the transport velocity is of the same order of magnitude as it was used for $\mathrm{SO_2}$ advection (this assumption is justified by similar depths of the atmospheric layers — around $1.5-2$ km (Warneck, 2000) and taking $4-6$ days as a typical value for $\mathcal{T}_{\mathrm{SO_4}}$ (Warneck, 2000; Surkova, 2002; Houghton et al., 2001, their Table 5.5), we estimate $L_{\mathrm{SO_4}} \sim 5 \times L_{\mathrm{SO_2}}$.

## 2.2 Horizontal transport

Taking into account just obtained estimates for $L_{\mathrm{SO_2}}$ and $L_{\mathrm{SO_4}}$, we may construct the transport scheme as follows.

At first, we solve the first Eq. (5). Because it is linear with respect to $B_{\mathrm{SO_2}}$, we may consider the model grid as an array of non–interacting sulphur sources numbered as $j = 1, 2, \ldots$. To reduce the computational burden, we consider only grid cells



with $E_{\mathrm{SO}_2}^{(j)} \geq E_{\mathrm{SO}_2,\min}$ and set $E_{\mathrm{SO}_2,\min}$ to sufficiently small, empirically chosen value (Table 1). At the source grid cell, the burden $B_{\mathrm{SO}_2}^{(j)}(\boldsymbol{\rho}_j)$ is calculated by using Eq. (6) with $Y = \mathrm{SO}_2$ and with $\gamma_{\mathrm{SO2}} = k_{\mathrm{SO2}}(\Delta x/u + \Delta y/v)$. Here $\boldsymbol{\rho}_j$ is the

horizontal coordinates of the grid cell corresponding to the source $j$, $E_{\mathrm{SO}_2}^{(j)} = E_{\mathrm{SO}_2}(\boldsymbol{\rho}_j)$, $\Delta y = \mathcal{R}_{\mathrm{E}}\Delta\phi$, $\Delta x = \mathcal{R}_{\mathrm{E}}\cos\phi\Delta\lambda$, $\mathcal{R}_{\mathrm{E}}$ is Earth radius, $\phi$ is latitude, $\Delta\phi$ and $\Delta\lambda$ are grid cell sizes in latitudinal and longitudinal directions, respectively.

The difference between $E_{\mathrm{SO}_2}^{(j)}$ and $k_{\mathrm{SO}_2}(\boldsymbol{\rho}_j)B_{\mathrm{SO}_2}^{(j)}(\boldsymbol{\rho}_j)$ is transported out of the source cell by advection:

$$F_{\mathrm{SO}_2,\mathrm{out}}^{(j)}(\boldsymbol{\rho}_j) = E_{\mathrm{SO}_2}^{(j)} - k_{\mathrm{SO}_2}B_{\mathrm{SO}_2}^{(j)}. \tag{9}$$

This flux is partitioned into zonal and meridional components in proportion to the corresponding wind component and to the

geometric size of the corresponding boundary of the cell:

$$
\begin{aligned}
F_{\mathrm{SO}_2,u}^{(j)} &= \frac{v\Delta y}{|u\Delta y| + |v\Delta x|} F_{\mathrm{SO}_2,\mathrm{out}}^{(j)}, \\
F_{\mathrm{SO}_2,v}^{(j)} &= \frac{v\Delta x}{|u\Delta y| + |v\Delta x|} F_{\mathrm{SO}_2,\mathrm{out}}^{(j)}.
\end{aligned}
\tag{10}
$$

The direction of each $F_{\mathrm{SO}_2,u}^{(j)}(\boldsymbol{\rho}_j)$ and $F_{\mathrm{SO}_2,v}^{(j)}(\boldsymbol{\rho}_j)$ is determined by the direction of zonal and meridional wind respectively.

Then we loop in the zonal direction and calculate $\mathrm{SO}_2$ burdens, related to the source $j$, as well as corresponding chemical and depositional losses, and fluxes out of the cell by using Eqs. (6), (9), and (10) (Fig. 1). In each cell $i$ included in this loop,

zonal flux from the previous zonal cell $F_{\mathrm{SO}_2,u}^{(j)}(\boldsymbol{\rho}_{i-1})$ is used in place of $E_{\mathrm{SO}_2}^{(j)}$, and $k_{\mathrm{SO}_2}(\boldsymbol{\rho}_j)$ is replaced by $k_{\mathrm{SO}_2}(\boldsymbol{\rho}_i)$. The loop is stopped in the cell with the number $I_{(j)}$, at which any of the conditions is met:

– either zonal wind $u(\boldsymbol{\rho}_{I_{(j)}})$ changes sign relative to $u(\boldsymbol{\rho}_j)$

– or the whole latitudinal circle is looped over.

In the stopping cell, $F_{\mathrm{SO}_2,\mathrm{out}}^{(j)}(\boldsymbol{\rho}_{I_{(j)}}) = F_{\mathrm{SO}_2,v}^{(j)}(\boldsymbol{\rho}_{I_{(j)}})$.

The $\mathrm{SO}_2$ mass, which is advected from the source cell and from the each of the looped–over cells, is transported to the respective neighbour cell either to the north or to the south depending on sign of $v(\boldsymbol{\rho}_i)$. No advection of sulphur dioxide mass from this meridional neighbour cell is allowed, and the $\mathrm{SO}_2$ burden is calculated assuming the balance between meridional mass inflow and chemical loss (Fig. 1).

At the next step, the $\mathrm{SO}_2$ burden in each grid cell is obtained by summing over all grid–cell sources:

$$B_{\mathrm{SO}_2,\mathrm{a}}(\boldsymbol{\rho}) = \sum_j B_{\mathrm{SO}_2}^{(j)}(\boldsymbol{\rho}). \tag{11}$$

This field still lacks any impact of diffusion. To represent impact of $A_{\mathrm{SO}_2}$, we smooth $B_{\mathrm{SO}_2,\mathrm{a}}$ by using the $n_{\mathrm{smo}} \times n_{\mathrm{smo}}$ rectangular window with weights, which are inversely proportional to $2^{-l^2}$, where $l$ is the distance between the centres of the given grid cell in the window and the central grid cell of the same windows. The weights sum to unity, and the result of the smoothing is put into the central grid cell of the window. Thus,

$$B_{\mathrm{SO}_2,\mathrm{smo}} = SMO\left(B_{\mathrm{SO}_2,\mathrm{a}}\right), \tag{12}$$





where $SMO$ is the just described smoothing operator. We set $n_{\mathrm{smo}}$ equal to 5 in the contemporary implementation.

It is easy to show that, by construction, for each grid–cell source $E_{\mathrm{SO_2}}^{(j)} = \sum_{i'} k_{\mathrm{SO_2}}(\boldsymbol{\rho}_{i'}) B_{\mathrm{SO_2}}(\boldsymbol{\rho}_{i'})$, where $i'$ stands for the set of cells over which the zonal loop is performed together with their meridional neighbours (or, in other words, the set of all coloured cells in Fig. 1). Thus, our advection scheme conserves mass up to the rounding errors. In turn, $B_{\mathrm{SO_2,a}}$ is also

constructed with the mass conservation. However, our smoothing procedure leads to slight violation of the mass conservation. We chose to recover this conservation by adjusting $B_{\mathrm{SO_2,smo}}$ with the scalar adjustment coefficient $\nu_{\mathrm{adj}}$:

$$
\begin{aligned}
B_{\mathrm{SO_2}} &= \nu_{\mathrm{adj}} B_{\mathrm{SO_2,a}}, \\
\nu_{\mathrm{adj}} &= \frac{\sum_{\mathrm{global}} E_{\mathrm{SO_2}}(\boldsymbol{\rho})}{\sum_{\mathrm{global}} k_{\mathrm{SO_2}}(\boldsymbol{\rho}) B_{\mathrm{SO_2,a}}(\boldsymbol{\rho}),}.
\end{aligned}
\tag{13}
$$

where "$\sum_{\mathrm{global}}$" stands for the area–weighted summation over all model grid cells. Such adjustment leads to the small (up to few per cent relative to the non–adjusted values) errors in calculated burdens, but allows to study the global sulphur budget.

A similar procedure is applied for $B_{\mathrm{SO_4}}$ calculation. At first, $SO_4$ source intensities are calculated from $SO_2$ burdens as specified in the first Eq. (4). Again, only points with $P_{\mathrm{SO_4}} \geq E_{\mathrm{SO_2,min}}$ are chosen to perform the calculations. Sulphates loss coefficients are calculated from the second Eq. (4). Then, we account for advection and diffusion of $SO_4$ in the same fashion as it is already done for $SO_2$.

At the final step, we calculate surface concentrations of sulphur dioxide and of sulphates from the calculated burdens em-

ploying Eq. (2). For this, we use the vertical scales $H_{\mathrm{SO_2}} = 1.2 \times 10^3$ m and $H_{\mathrm{SO_4}} = 1.8 \times 10^3$ m (Jaenicke, 1993; Warneck, 2000).

The ChAP data flow is summarised in Fig. 2.

## 2.3   Parametrisation of chemical sources and sinks

We assume that $k_{\mathrm{in-cl}}$ is proportional to cloud fraction $c$ and cloud water path and, in addition, depends on temperature because

of the respective dependence of the involved reaction rate constants. As a result, we chose to use

$$
k_{\mathrm{in-cl}} = k_{\mathrm{in-cl,0}} \cdot e^{\alpha_{\mathrm{in-cl}}(T-T_0)} \cdot c^{\beta_{\mathrm{in-cl}}},
\tag{14}
$$

where $k_{\mathrm{in-cl,0}}$, $\alpha_{\mathrm{in-cl}}$, and $\beta_{\mathrm{in-cl}}$ are constants, $T_0 = 288$ K. In this equation, dependence on cloud parameters is constructed by assuming that most of the $SO_2$ oxidation occurs in the cloud–covered part of the model grid cell and taking into account that at the coarse spatial and time scale the cloud water path depends on $c$ approximately as a power function (Eliseev et al.,

2013). The oxidation rate $k_{\mathrm{in-cl}}$ dependence on temperature is uncertain as well because this conversion is not a single–step reaction and depends on solubilities of sulphur substances in water and on the rate of the $SO_2$ oxidation by peroxide radical. Therefore, it is difficult to relate $\alpha_{\mathrm{in-cl}}$ directly to the activation energies of these reactions. However, such activation energies are able to provide an order–of–magnitude estimate for value of this coefficient. For instance, the activation energy value for reaction $HSO_3 + H_2O_2$ as listed in Table 1 of (Barth et al., 2000) for typical lower tropospheric temperatures corresponds to

$\alpha_{\mathrm{in-cl}} = 0.05$ K$^{-1}$. We use this value as a guide below.





Recall that $k_{\mathrm{gas}} = 0$ because of $R_{\mathrm{gas}} = 0$ (Sect. 2.1). Thus, the production of sulphates, $R_{\mathrm{SO_4,prod}} \equiv R_{\mathrm{in-cl}}$. We will discuss this limitation below (Sect. 6).

We set $k_{\mathrm{SO_2,dry}}$ and $k_{\mathrm{SO_4,dry}}$ to constant values (Table 1). Because sulphates wet deposition should depend on precipitation rate $p$ and this dependence is expected to saturate somewhat in a limiting case of very strong precipitation, after some trial–
and–error procedure we chose

$$k_{\mathrm{SO_4,wet}} = k_{\mathrm{SO_4,wet,0}} \times \arctan{(p/p_0)}, \tag{15}$$

where $k_{\mathrm{SO_4,wet,0}}$ and $p_0$ are constants.

## 3   Simulations setup

We ran our model for 1850-2000 with the $SO_2$ emissions data from the CMIP5 (Coupled Models Intercomparison Project,
phase 5) 'historical' database (Lamarque et al., 2010) (see also Fig. S3). This database lacks the global gridded data on $B_{\mathrm{SO_2}}$ but provides the data on $B_{\mathrm{SO_4}}$ (Lamarque et al., 2013b). The data on sulphate burden were used to evaluate the performance of our scheme. Both emission and burden data are available as time slices with a step of 10 yr.

The CMIP5 data are recently superseded by the CMIP6 (Coupled Models Intercomparison Project, phase 6) data sets (Hoesly et al., 2018; Turnock et al., 2020). However, because the CMIP5 data are sufficient to validate our scheme, and because we
expect that this scheme would need some (but not major) retuning when it is implemented into an Earth System Model, we limit our calculations in the present paper with the CMIP5 data. We postpone the task to run our scheme with the CMIP6 emissions for the next stage — when our scheme is implemented into EMIC.

In our calculations, we neglect dimethyl sulphide emissions from the ocean, which is an important source of the sulphur dioxide in the marine atmosphere (Warneck, 2000; Surkova, 2002). We do it mostly because they are not available in the
CMIP5 forcing data (see https://tntcat.iiasa.ac.at/RcpDb/). Moreover, we neglect other, more minor sulphur sources such as volcanos and the terrestrial biosphere. Thus, we assume that natural sources did not change since year 1850, which is in the CMIP5 protocol is considered as a pre–industrial year. In addition, we note that an implementation of the natural sources for the considered here sulphur compounds, while certainly important, would complicate our scheme. Again, we postponed this task for future work. Therefore, we compared the given year our simulation with the difference of the CMIP5 data for this year
from the respective data for year 1850.

In addition, we neglected the direct anthropogenic $SO_4$ emissions into the atmosphere which contribution to the sulphur budget is generally small (Houghton et al., 2001, their Table 5.5). However, a possibility to account for these emissions is already coded in ChAP and may be used in future simulations.

We use the monthly mean ERA–Interim data (Dee et al., 2011) averaged over 1979–2015 to force our scheme. This set
up neglects dependence of meteorological variables on time, and, therefore, respective dependencies of spieces advection. In addition, this approach ignores interannual changes of temperature in Eq. (14). However, a similar neglect is embedded into the construction of the CMIP5 $SO_4$ burdens (Lamarque et al., 2013b). Thus, our approach even makes the evaluation of our scheme more straightforward.





All forcing fields were interpolated on a common grid with $40 \times 60$ latitude–longitude grid corresponding to the horizontal
resolution of $4.5^{\circ} \times 6.0^{\circ}$. This resolution was chosen to correspond the IAPRAS-MSU EMIC (Eliseev et al., 2007; Mokhov
and Eliseev, 2012; Eliseev et al., 2014), which is considered as a primary hosting model for our scheme. This resolution is also
quite similar to the resolution employed in other Earth system models of intermediate complexity (Eby et al., 2013).

## 4   Tuning procedure

To tune our scheme, we follow the procedure which is similar to that used by Eliseev et al. (2013). At first, we tune it manually
to achieve a first-guess, reasonable performance. At the next stage, we sample the first 7 parameters listed in Table 1 in the
predetermined intervals. The sampling was done by using the Latin hypercube sampling (McKay et al., 1979; Stein, 1987) to
insure that the ensemble statistics is unbiased. The sample length is $K = 5,000$.

For each individual simulation $1 \le k \le K$ (in this section, $k$ is a simulation label arather than chemical or depositional loss
coefficient) and for each calender month $m$, the skill score in each grid cell $\boldsymbol{\rho}$ is defined based on the ratio $\eta(\boldsymbol{\rho})$ of the modelled
$SO_4$ burden per unit area to the observed one, $B_{SO_4,o}$:

$$s_{k,m}(\boldsymbol{\rho}) \propto \exp\left\{ -\frac{[\eta(\boldsymbol{\rho})-1]^2}{2} \right\}. \tag{16}$$

from which the area–weighted global skill score $\tilde{s}_{k,m}$ is constructed. Finally, skill score for simulation $k$ is calculated by
multiplying the respective skill scores for boreal winter ('win': from December to January, DJF) and summer ('sum': from
June to August, JJA):

$$S_k \propto s_{k,\text{win}} \cdot s_{k,\text{sum}}, \tag{17}$$

We standardise skill scores $S_k$ by applying a condition that they should sum to unity

$$\sum_k S_k = 1. \tag{18}$$

Upon completing model runs with each parameter set from this sample, we selected only those runs which fulfil the require-
ments $S_k \ge 0.06$, $0.8 \le R_{SO_4,\text{prod}}/D_{SO_2,\text{dry}} \le 1.2$ and $\mathcal{T}_{SO_4} < 7$ days. The first requirement is based on the observation that
the maximum value of $S_k$ is 0.7113, so we choose the simulations which are close to the optimal. The second requirement
arises from simulations reported in (Warneck, 2000; Surkova, 2002; Houghton et al., 2001, their Table 5.5), in which sulphur
dioxide emissions were almost equipartitioned between production of sulphates and $SO_2$ deposition. The third requirement is
based on typical lifetimes of sulphates in the atmosphere. While it looks redundant taking into account our calculation of $s_{k,m}$,
it is necessary to avoid an overfitting of the observed fields in the regions of small sulphate burdens, which are not so important
for climate and ecological applications. Such overfitting in our simualtions tends to bias the model with underestimated $SO_4$
production (this, in addition, motivated us to implement the requirement on $R_{SO_4,\text{prod}}/D_{SO_2,\text{dry}}$) and deposition of sulphates,
despite of the reasonable $SO_4$ burden.





As a result, 40 simulations were considered as being close to the optimal. The means of parameters over these simulations were considered as a tuned parameter set, and their standard deviations were considered as a measure of uncertainty for these

parameters.

## 5    Perfomance

The tuned parameter values and their uncertainties are listed in Table 1. Below, only the simulations with the tuned set of parameters is discussed.

We assessed the performance of our tuned scheme by comparing it to

– the original CMIP5 data (to avoid a circular reasoning, we highlight that it is an evaluation of our tuning procedure rather then of the implemented physics);

       – ACCMIP phase II simulations (Lamarque et al., 2013a; Myhre et al., 2013), which were performed both for the preindustrial and for the present day emissions of aerosols and their precursors to the atmosphere (thus, the difference between these simulations is an analogue to our anthropogenic–only simulations). The caveat, however, is due to difference in

270       prescribed $SO_2$ emissions between our simulations and the ACCMIP protocol (Table 2).

In addition, we use two datasets based on the assimilation of the available measurements into the chemical–transport models: the Copernicus Atmosphere Monitoring Service (CAMS), (Inness et al., 2019) and the Meteorological Synthesizing Centre–West of the European Monitoring and Evaluation Programme (EMEP MSC-W), (Simpson et al., 2012), see supplementary Figs. S4–S7. These data can not be used for direct comparison to our simulations because they are forced by both anthro-

pogenic and natural emissions into the atmosphere, and the impact of the latter emissions can not be factored out because no preindustrial simulations are available. However, we can compare our simulations with both datasets in the regions of strong anthropogenic sulphate pollution of the atmosphere, such as Europe, south-east Asia or North America (Chin et al., 2000) assuming that here anthropogenic sulphur load dominates over the natural one. Similar intercomparison may be made with individual model simulations summarised in Table 5.5 of IPCC TAR (Houghton et al., 2001). We note that, in contrast to the

CMIP5 and ACCMIP datasets, CAMS and EMEP MSC-W were prepared by using the meteorology which changes from year to year making makes such comparison less straightforward. The individual model simulations in the IPCC TAR Table 5.5 were performed in a stationary fashion, with the year-to-year changes in meteorology only due to internal model variability. Therefore, we can consider them as also being ran with an 'almost constant' meteorological fields, which makes our comparison with these simulations more straightforward.

Below we first discuss the performance of our scheme with respect to simulation of $SO_2$, because it is independent from the $SO_4$ performance. Then we proceed with a similar discussion of $SO_4$ simulation which, in contrast, depends on the calculated $SO_2$ burden. We note that the maximum anthropogenic $SO_2$ emissions into the atmosphere in the CMIP5 data correspond to the 1980 time slice. However, because the model results for year 1990 are quite similar to those for year 1980, and because





the most data are available since year 2000, we use the time slice for year 1990 as a primary model output to compare to the
existing data. In such cases, we use the year 2000 time slice as a primary source for comparison.

## 5.1 Simulation of $SO_2$ burden and near-surface concentration

At the global scale, about half of the $SO_2$ emissions in our model are consumed by the chemical $SO_4$ production in the
atmosphere, and another half is deposited to the surface in the form of sulphur dioxide (Table 2). This fractions are within the
ranges reported in IPCC TAR Table 5.5 ($SO_2$ deposition: from 18 to 56% of the prescribed emissio rate with ensemble mean
and ensemble standard deviations $42 \pm 12\%$; $SO_4$ production: correspondingly from 42 to 74%, $57 \pm 12\%$). Anthropogenic
sulphur dioxide burden monotonically increases until 1980, reaches $\approx 0.2\,\text{TgS}$ in 1970–1990 and drops to $0.16\,\text{TgS}$ in 2000
(Table 2, Fig. 3a). These burdens are in the lower range of the IPCC TAR Table 5.5-derived values (Table 2). The sulphur
dioxide lifetime in the atmosphere in the model is close to $1.1\,\text{day}$. This value is within the respective lifetimes reported in
IPCC TAR Table 5.5. This lifetime in our simulations changes non-systematically between different time slices with standard
deviation of $0.02\,\text{day}$. Such variations are caused by the employed smoothing procedure.

For the 1990 and 2000 time slices, annual mean sulphur dioxide burden exhibits maxima in the regions of the strong anthro-
pogenic pollution – Europe, south-east Asia, and eastern North America, where $B_{SO_2}$ is typically larger than $2\,\text{mgS}\,\text{m}^{-2}$, and
in some grid cells it is in excess of $5\,\text{mgS}\,\text{m}^{-2}$ (Fig. 3b-e). Smaller maxima of $B_{SO_2}$ with typical values $1-2\,\text{mgS}\,\text{m}^{-2}$ are
found in south Africa and in the western part of South America. In 1990 (as well as in previous years) $SO_2$ in Europe is larger
than in south-east Asia, while in 2000 the regional maximum in south-east Asia is larger than in other regions. This is quite
expected based on the regional differences in sulphur dioxide emissions (Fig. S3; the emissions in years 1980 and 1990 are
similar to each other).

Geographical distribution of near-surface $SO_2$ concentration basically follows that of the sulphur dioxide total column
burden (Fig. 4). Again, in the anthropogenically polluted regions, $q_{SO_2,s}$ in the last decades of the 20th century is above
$2\,\mu\text{gS}\,\text{m}^{-3}$, and in Europe until the 1990 time slice it is larger than $5\,\mu\text{gS}\,\text{m}^{-3}$. In south Africa and in the west of South
America source regions, near-surface $SO_2$ concentration is from 1 to $2\,\mu\text{gS}\,\text{m}^{-3}$.

The time slice for year 2000 in the model reasonably agrees with the CAMS data for 2003-2010 in the above-mentioned
regions of strong pollution for both total column burden and near-surface concentration of sulphur dioxide (Figs. S4 and S5).
Nonetheless, one notes some overestimation of both variables in Europe and some underestimate in south-east Asia. In Europe,
the ChAP-simulated $q_{SO_2,s}$ also agrees with the EMEP MSC-W data for 2000-2005 (Fig. S6). The larger discrepancy of our
simulations in Europe from the CAMS data than from the EMEP MSC-W data is at least partly explained by difference in cov-
ered period between the CAMS and EMEP MSC-W datasets. Namely, provided that aerosol emissions in Europe continuously
decrease in the early 21st century, one may expect that the mean over 2000-2006 is closer to the time slice 2000 compared
to the 2003-2020 average. We also note that the ChAP-simulated values in central Europe in 1990 generally agree with the
older EMEP data for mid-1990's as summarised by Semenov et al. (1998). Moreover, in the regions of strong pollution, our
burden for year 1990 is similar to that simulated with the NCAR CCM (Barth et al., 2000), GISS (Chin et al., 2000) and





CCCMA (Lohmann et al., 1999) models. Near-surface sulphur dioxide concentrations are comparable to those simulated with the IMAGES (Pham et al., 1995) and GISS (Chin et al., 1996) models.

## 5.2 Simulation of $SO_4$ burden and near-surface concentration

Similar to it was for $SO_2$, the total column burden of anthropogenic sulphates monotonically increases until 1980, reaches $\approx 0.4\,\mathrm{TgS}$ in 1970–1990 and drops to $0.32\,\mathrm{TgS}$ in year 2000 (Table 2, Fig. 5a). These values are only slightly smaller than the corresponding values from the CMIP5 database. In addition, the value for time slice 2000 is close the range obtained in the respective ACCMIP exercise (Myhre et al., 2013), while they are in the lower part of this range. Our simulated total column burden of sulphates in year 1990 is also within the IPCC TAR estimates, albeit again in its lower part.

Similar to that it was for the sulphur dioxide, the sulphates lifetime in the atmosphere in the model changes non-systematically between different time slices with mean of $4.8\,\mathrm{days}$ and standard deviation of $0.2\,\mathrm{days}$. Again, these variations are caused by the employed smoothing procedure. The value of $\mathcal{T}_{SO_4}$ for the year 1990 time slice is within the respective lifetimes reported in individual model simulations (Houghton et al., 2001, Table 5.5). In addition, the modelled $\mathcal{T}_{SO_4}$ is in agreement with the recent AeroCom phase III simulations for year 2100, which lead to the range from 2.6 to 7.0 days with the ensemble mean of

4.9 days and ensemble standard deviation of 1.6 days (Gliß et al., 2021).

As it is expected, the principal regions of the atmospheric pollution by sulphates are similar to those obtained for sulphur dioxide in the previous Section. However, because of several-fold larger $\mathcal{T}_{SO_4}$ relative to $\mathcal{T}_{SO_2}$, sulphates are transported at larger distances in comparison to sulphur dioxide, and the individual source regions become 'visually connected' on maps. In Europe, south-east Asia, and in south-east of North America, $B_{SO_4}$ from 1970's till the end of simulation is in excess of

$2\,\mathrm{mgS\,m^{-2}}$, and it is above $5\,\mathrm{mgS\,m^{-2}}$ for large areas in Europe and in south-east Asia during period of the strongest $SO_4$ anthropogenic loading – in 1980 and in 1990 (Figs. 5-7). In the smaller in magnitude spatial maximum in south Africa, this variable typically in 1970-2000 amounts $1-2\,\mathrm{mgS\,m^{-2}}$. This is somewhat in contrast to another spatial maximum in South America – despite sulphur dioxide burdens per unit area in South America and in south Africa are similar in 1970-2000 in our simulations, the respective $SO_4$ burden in South America is closer to the European, south-east Asian, and North American

ones than to that in south Africa. This difference is caused by very small zonal velocity in the South American source region (Fig. S2), which leads to very small transport of sulphates out of this region. In turn, the effect of horizontal transport is less pronounced for sulphur dioxide owing to the difference between $\mathcal{T}_{SO_4}$ and $\mathcal{T}_{SO_2}$.

Geographic distribution of the modelled $B_{SO_4}$, as a whole, similar to that in the CMIP5 database (Figs. 5-7). However, for the period of the strongest $SO_2$ atmospheric emissions, burden of sulphates in Europe is systematically overestimated by

our model, especially in winter. For summer, the correspondence of the ChAP-simulated and CMIP5 burdens is better. The agreement of $SO_4$ burden per unit area in south-east Asia depends on season: in winter our model overestimates the sulphates burden in this region, and in summer it underestimates it, but to a lesser extent than in winter. Mutual compensation between the model biases in different seasons lead to overall reasonable simulation of sulphate burden per unit area in this source region. The magnitude of $B_{SO_4}$ in the North American source regions is basically correct, but during summer the maximum in this





region is shifted to the west. The latter feature is not exhibited in winter. The $B_{SO_4}$ magnitudes in the source regions in the Southern Hemisphere are overestimated for the whole year.

Compared to the CAMS reanalysis (Fig. S4) in major source regions, our model overestimates sulphates burden per unit area in Europe with the larger discrepancy in winter then in summer. The $B_{SO_4}$ pattern in south-east Asian source region is underestimated – this differs to that obtained in the comparison between our ChAP simulation and the CMIP5 database.

The latter difference, at least partly, is due to difference in covering periods (recall, that CAMS is for 2003-2010). Again, the magnitude of the $SO_4$ burden in the North American source region is realistic in ChAP, but now we see that even the location of maximum is correct. Thus, our previous conclusion about this location is points to some possible shortcomings in the CMIP5 dataset. In the Southern Hemisphere, our model overestimates sulphate burdens per unit area in both south African and South American source regions. In addition, $B_{SO_4}$ in the Northern Hemisphere source regions is similar to those reported in the

simulations with the NCAR CCM (Barth et al., 2000; Rasch et al., 2000), ECHAM (Feichter et al., 1996; Roelofs et al., 1998), GISS (Chin et al., 2000), and CCCMA (Lohmann et al., 1999) models.

Geographic distribution of near-surface $SO_4$ concentration, $q_{SO_4,s}$, follows the corresponding distribution of $B_{SO_4}$ (Fig. 8). Identically to that it was for sulphur dioxide, this is a direct consequence of Eq. (2). In the anthropogenically polluted regions, $q_{SO_4,s}$ in the last decades of the 20th century is above $2\,\mu gS\,m^{-3}$, and in Europe during summer 1990 it is larger than

$5\,\mu gS\,m^{-3}$. In south Africa and in the west of South America, near-surface $SO_2$ concentration is typically above $0.5\,\mu gS\,m^{-3}$. The simulated near-surface $SO_4$ concentration are generally similar to those with the IMAGES (Pham et al., 1995) and GISS (Chin et al., 1996) models.

The modelled $q_{SO_4,s}$ in year 2000 in the principal source regions reasonably corresponds to the CAMS data for 2003-2010 (Fig. S5), but with an overestimate in Europe in summer and an underestimate in south-east Asia throughout the year.

In Europe, our time slice for year 2000 systematically exhibits larger near-surface concentration of sulphates relative to the EMEP MSC-W data for 2000-2005 (Fig. S6).

### 5.3 Simulation of annual $SO_x$ deposition

Owing to the mass conservation, the global $SO_x$ deposition in the model is equal to the applied sulphur dioxide emissions. Depending on time slice, dry $SO_x$ deposition $D_{SO_x,dry} = D_{SO_2,dry} + D_{SO_4,dry}$ explains from 55 to 59% of the total $SO_x$

deposition (mostly in the form of $SO_2$), and wet deposition $D_{SO_x,wet} = D_{SO_4,wet}$ explains another $41-45\%$ (only in the $SO_4$ form by construction) (Table 2, Fig. 9a). The contribution of wet $SO_x$ deposition in 1980 and 2000 is also similar to that obtained from ACCMIP (46% and 51%, respectively, Lamarque et al., 2013a) and is within the ranges reported in Table 5.5 of(Houghton et al., 2001) for year 1990 (from 37 to 64% with mean of 47% and median of 45%).

Geographic distribution of the total $SO_x$ deposition $D_{SO_x} = D_{SO_x,wet} + D_{SO_x,dry}$ is very close to the sulphur dioxide

emissions in a given year (cf. Fig. 9a with Fig. S3d and Fig. S7a with Fig. S3e). For year 2000, total deposition is above $1\,MgS\,m^{-2}\,yr^{-1}$ in the cores of the Northern Hemisphere source region and is above $0.2\,MgS\,m^{-2}\,yr^{-1}$ in the respective Southern Hemisphere source cores (Fig. 9a). In year 1980 (and in 1990 as well; not shown) the corresponding values in Europe are even larger, $> 2\,MgS\,m^{-2}\,yr^{-1}$. For both years there is a quite close agreement of the modelled $D_{SO_x}$ with the ACCMIP





data (Figs. 9b, c and S7a, b). Again, this is a validation for our numerics and for our code rather than for the implemented
physics, because total $SO_x$ deposition near any source region is controlled by prescribed emissions and by prescribed winds.

More stringent test of the implemented physics is a subdivision of total $SO_x$ deposition into wet and dry ones (Figs. 9d-g
and S7c-f). This shows that ChAP generally overestimates wet deposition and underestimates dry one relative to the ACCMIP
simulations. This is not visible in the global numbers (Table 2, Fig. 9a) because of differences in extent of the regions in
which 'substantial' (say, $\geq 0.1\,\mathrm{MgS\,m^{-2}\,yr^{-1}}$ in Figs. 9d-g and S7c-f) deposition occurs. However, in Europe an agreement
is markedly better with the EMEP MSC-W data (Fig. S8). The ChAP-simualted wet $SO_x$ deposition in year 1990 (which is
rather similar to year 1980) is also in a general agreement with the simulations with the MOGUNTIA (Langner and Rodhe,
1991), IMAGES (Pham et al., 1995), and GISS (Koch et al., 1999) models.

## 6   Limitations of the current version of the scheme and future prospects

It was demonstrated in the previous Section that, despite of its apparent simplicity, ChAP-1.0 is able to reproduce gross
characteristics of the tropospheric sulphur cycle for late 19th and the whole 20th century. However, our model has inherent
limitations, which have to be discussed together with figuring out the way to extend and improve ChAP.

First of all, in the contemporary version of ChAP does not implement any scheme for contribution of dimethyl sulphide
(DMS) and other minor atmospheric sulphur species to chemical production sulphur dioxide. DMS emissions are basically
biogenic and mostly limited to the ocean. According to the existing estimates, atmospheric DMS burden changes no more by
few per cent even under strong climate changes as assessed in (Houghton et al., 2001, Sect. 5.5.2.1) and further reported by
Bopp et al. (2003) and by Kloster et al. (2007). Thus, given the present-day DMS source strength up to $28\,\mathrm{TgS/yr}$ (Lana et al.,
2011; Galí et al., 2018; Wang et al., 2020), DMS lifetime in the atmosphere from 1 to 3 days and its complete conversion to
$SO_2$ (Table 2 in Koch et al., 1999), such increase would change sulphur dioxide and suphate burdens in the troposphere mostly
over oceans. We plan to implement this source into our scheme in future.

ChAP also misses other natural sulphur sources into the atmosphere: non-eruptive volcanic (the present-day strength is
$23\pm2\,\mathrm{TgS/yr}$ (Carn et al., 2017); $SO_2$ release from volcanic eruptions is of order of magnitude smaller and is partly loaded into
the stratosphere than into the troposphere (https://disc.gsfc.nasa.gov/datasets/MSVOLSO2L4_3)) and from biomass burning
(correspondingly, $\approx 1.2\,\mathrm{TgS/yr}$ (van der Werf et al., 2017); this source is partly anthropogenic). However, the former source
may be considered as constant in time. The latter source, even it triples following other emissions (which may occur under
high-$CO_2$ anthropogenic scenario (Eliseev et al., 2014)), would not change the global sulphur budget markedly, albeit may
be important at regional level. These sources may be readily added to our scheme as contributions to $E_{SO_2}$. We do plan to
implement them in future provided that a hosting EMIC able to simulate natural fires (Sitch et al., 2005; Eliseev and Mokhov,
2011; Eliseev, 2011; Eliseev et al., 2014, 2017).

The model knows nothing about availability of oxidants (OH, $HO_2$, and $O_3$). This is apparently equivalent to the assumption
that these oxidants are always abundant. Partly this assumption is ameliorated by relating the atmospheric sulphur dioxide
oxidation rate to cloud fraction, which is a characteristics of the atmospheric hydrological cycle. Nevertheless, some hint that





this assumption should be relaxed may be obtained from mutual comparison of global burdens of $SO_2$ and $SO_4$: while both are in the lower half of the IPCC TAR range, the former is closer to the corresponding median (Fig. 3a) in comparison to the latter (Fig. 5a). It may be possible to implement stationary equations for hydroxyl- and peroxide-radicals owing to their very

short (e.g., no more than few seconds in the lower troposphere (Lelieveld et al., 2016)). However, such stationary assumption is likely to be problematic for ozone with its typical lifetime of several weeks (Young et al., 2013).

Our horizontal transport solver is very simplistic and is able to provide more or less realistic results only for species with lifetimes up to few days (Sect. 2.1 and 2.2). Taking into account the estimated horizontal advection length scales, $L_{SO_2}$ and $L_{SO_2}$ one sees that our approach is well justified for zonal advection (because, typically, a large number of grid cells is included

into a zonal loop in Sect. 2.2) but becomes more suspicious for the meridional one. This shortcoming, however, is somewhat compensated by our rather large value of $n_{smo}$, which in midlatitudes corresponds to the length scale of $(1-2) \times 10^3$ km — this is pretty comparable to $L_{SO_4}$. In future we plan either to improve our solver or just to combine cells in meridional direction to make possible transporting species over longer distances owing to decreased meridional resolution.

Another transport-solver related issue is due to implemented smoothing procedure. This implementation is reasoned by

neglect of synoptic scales in our transport routine – we use only monthly mean winds. While ChAP is formally linear with respect to horizontal winds (Eq. (5), and, therefore allows averaging over synoptic-scale motions, possible time correlations between synoptic-scale variations of winds and of pollutants burdens make the underlying processes nonlinear (Saltzman, 1978; Branscome, 1983; Petoukhov et al., 2008; Coumou et al., 2011). Further, one may argue that the corresponding mixing length (thus, $n_{smo}$) could be made dependent on synoptic-scale kinetic energy (Branscome, 1983; Coumou et al., 2011). At the

time being, $n_{smo}$ is a parameter of the scheme, but in future it could become depending on large-scale atmospheric state (e.g., on the state with time and space scales larger than synoptic ones).

In our scheme we neglected wet deposition of sulphur dioxide. This was done based on the synthesis of simulations listed in Table 5.5 of IPCC TAR (Houghton et al., 2001), in which wet deposition of $SO_2$ explained no more than 15% of sulphur dioxide budget. Only in CCM1-GRANTOUR (Chuang et al., 1997) and the earlier version of GOCART (Chin et al., 1996) this

contribution is from 15 to 20%. While the upper-end values from these papers are not negligible, $D_{SO_2,wet}$ is still neglected. Its implementation would probably improve regional performance of our scheme. We acknowledge the neglect of $D_{SO_2,wet}$ as a limitation of our scheme.

In addition, is is necessary to highlight that dry deposition of sulphates is still included in our scheme, despite its contribution to the $SO_4$ budget is also not so important relative to the $SO_2$ wet deposition at the global scale. For instance, for the IPCC

TAR ensemble this contribution is $\leq 25\%$. The reason for keeping $D_{SO_4,wet}$ is mostly numeric: such background deposition avoids a division by zero in regions with very small precipitation rate.

Gas-phase oxidation of sulphur dioxide is also formally neglected in the current version of ChAP. Depending on the model, gas-phase may be or may be not an important process in converting $SO_2$ to $SO_4$ (see, e.g., Table 2 in (Koch et al., 1999)). However, our scheme still accounts implicitly for for gas-phase oxidation of sulphur dioxide because the total sulphate production

is optimised rather than only its in-cloud part. The reason for the latter is due to the major gas-phase oxidant (hydroxyl radical)




which is also produced in the atmospheric hydrological cycle-related pathways. We neglect the sulphur dioxide oxidation by ozone as well. This unlikely to be covered by any tuning of Eq. (14), and we acknowledge it as a limitation for ChAP.

More subtle issue is due to implementation of total cloud fraction in Eq. (14) for in-cloud oxidation rate. Because most sulphur conversion occurs in the lower half of the troposphere, one may argue that low cloud fraction would be a better
predictor for this oxidation rate. We tried this option during development of ChAP and found no marked differences (apart somewhat different optimal values of parameters listed in Table 1). Thus, to be in line with the contemporary generation of EMICs, which mostly do not provide cloud fractions for different layers, we kept total cloud fraction in the input to our scheme instead of low cloud fraction.

Another way of improving the calculation of near-surface concentrations of sulphur species is to account for regional differ-
ences of $H_{SO_2}$ and $H_{SO_4}$. This may be done be relating its values to large-scale values of the planetary boundary layer depth (thus, to vertical mixing inside this layer). We postpone this as a possible future extension of our scheme.

Performance of the scheme was mainly tested against the CMIP5 data and the ACCMIP simulations. The basic reason for limiting validation to these datasets is due to our neglect of the natural sources of sulphur emissions into the atmosphere. This limitation is somewhat relaxed in the present paper by comparing to the CAMS and EMEP output and to individual
simulations in regions of strong anthropogenic pollution into the atmosphere. Our neglect of natural sulphur sources is also one of the reasons to exclude direct measurements of sulphur burdens (e.g., Aas et al., 2019). Another reason for this exclusion is due to possible complications owing to local features which are likely present in these direct, point-scale measurements. A meaningful use of such data would need their stratification into background and polluted stations and factoring out such local-scale features, which is beyond the scope of the present study. Exclusion of such point-scale measurements is somewhat
ameliorated by their assimilation into CAMS and EMEP. In addition, in future exercises, we plan to replace the CMIP5 forcing by the CMIP6 one (Hoesly et al., 2018; Turnock et al., 2020).

A related issue is due to our use of the ERA cloud and precipitation fields rather than those based on direct measurements. For instance, arguably more reliable cloud fractions may be prescribed from the A-Train satellite observations (Minnis et al., 2011; Frey et al., 2008). Correspondingly, precipitation rate could be derived from the GPCP (Global Precipitation Climatology
Project) data (an update from Huffman et al., 2009). Marked differences of the ERA-Interim fields from these data are documented, for instance, by Dolinar et al. (2016), Stengel et al. (2018), and by Nogueira (2020). For instance, the underestimated rainfall rate in Europe in ERA-Interim (Nogueira, 2020) region may be the reason of our overestimate of $B_{SO_4}$ in this region, while the corresponding excessive precipitation rate in the Asian monsoon region could contribute to the underpredicted sulphate burden over south-east Asia. However, we prefer to keep the ERA-Interim cloud fractions and precipitation as forcing
fields in our tuning exercise because they are at least dynamically consistent with other forcing fields.

Finally, choice of skill scores is always subjective in the tuning exercises like ours. We reported only one version of skill scores (Eq. (16)). However, we tried another skill scores as well, e.g., either based on the root-mean-square errors (RMSE) in simulations or by limiting the skill scores calculations only to the regions with sufficiently large $SO_2$ and $SO_4$ burdens. The first option (RMSE skill score) provided rather unrobust results. The second options did not resulted to much improved
simulation with respect to that reported in Sect. 5. Therefore, we decided to use the skill score as figured in Eq. (16).





# 7  Conclusions

A stationary, computationally efficient scheme, ChAP-1.0 (Chemical and Aerosol Processes, version 1.0) for the sulphur cycle in the troposphere is developed. This scheme is designed to be implemented into Earth system models of intermediate complexity (EMICs). The scheme accounts for sulphur dioxide emissions into the atmosphere, its deposition to the surface,
oxidation to sulphates, and dry and wet deposition of sulphates on the surface. Horizontal transport of sulphur compounds in the atmosphere is tackled by representing model grid cells as non-interacting sources of particular sulphur species. The calculations with the scheme are performed forced by anthropogenic emissions of sulphur dioxide into the atmosphere for 1850-2000 adopted from the CMIP5 dataset and by the ERA-Interim meteorology. This setup assumes that natural sources of sulphur into the atmosphere remain unchanged during this period.

The ChAP output are compared to changes of the tropospheric sulphur cycle simulations: with the CMIP5 $B_{SO_4}$ data, with the IPCC TAR ensemble, and with the ACCMIP phase II simulations. In addition, in regions of strong anthropogenic sulphur pollution, ChAP results are compared to other data, such as the CAMS reanalysis, the EMEP MSC-W output, and with individual model simulations. Our model reasonably reproduces characteristics of the tropospheric sulphur cycle known from these information sources. In particular, in 1980 and 1990, when the global anthropogenic emission of sulphur is at the
maximum, global atmospheric burdens of $SO_2$ and $SO_4$ account, correspondingly, $0.2$ TgS and $0.4$ TgS. In our scheme, about half of the emitted sulphur dioxide is deposited to the surface and the rest in oxidised into sulphates. In turn, sulphates are mostly removed from the atmosphere by wet deposition. The lifetime of the sulphur dioxide and sulphates in the atmosphere is close to, respectively, 1 day and 5 days. The differences between our simulations, on one hand, and the CAMS and EMEP MSC-W datasets, on the other, are partly (but likely far from completely) explained by the differences in time intervals covered
by our simualtions and by these datasets.

We acknowledged the following major limitations of the contemporary version of ChAP:

– Omission of natural and, partly, of some anthropogenic sulphur emissions into the troposphere;

– Neglect of $SO_2$ wet deposition and (partly) of its gas-phase oxidation;

– Very indirect relationship between the intensity of sulphur dioxide oxidation rate and the amount of major oxidants;

– Simplifications in the transport solver.

We plan to relax these limitations during future development of our scheme.

Despite its simplicity, our scheme is able to reproduce gross characteristics of the tropospheric sulphur cycle during the historical period. Thus, it may be successfully used to simulate anthropogenic sulphur pollution in the atmosphere at coarse spatial and time scales. At next stage, we are going to implement it into EMIC and reproduce direct radiative effect of sulphates
on climate, their respective indirect (cloud- and precipitation-related) effects, as well as an impact of sulphur compounds on the terrestrial carbon cycle.



*Code and data availability.* The Fortran code for ChAP as well as all the data used in this paper are available at the ZENODO repository via https://doi.org/10.5281/zenodo.4513909.

*Author contributions.* A.V.E. designed the scheme, performed validation runs and wrote the first draft of the manuscript. R.D.G. contributed
by comparing the simulations with the EMEP and CAMS data. A.V.T. prepared the data to run the scheme. All authors contributed to the manuscript revisions after the first draft.

*Competing interests.* The authors declare that they have no conflict of interest.

*Acknowledgements.* The work is supported by the Russian Science Foundation grant 20-62-46056. Authors are grateful for J.-F. Lamarque for providing the data published in his and co-authors' (2013) paper.





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





## List of Figures





**Table 1.** Numerical parameters of ChAP. The symbol '—' in column 'LHS interval' indicates that this parameter was not sampled during the tuning procedure, and the respective number from column 'calibrated value' was used throughout the paper.

| parameter and units | equation | code name | LHS interval | calibrated value |
|---|---|---|---|---|
| $k_{\mathrm{in-cl},0}$, s$^{-1}$ | (14) | SO4PRODCOEF | $(0.2-5.0) \times 10^{-5}$ | $(3.0 \pm 1.2) \times 10^{-5}$ |
| $\alpha_{\mathrm{in-cl}}$, K$^{-1}$ | (14) | SO2SO4ALPHA | $0-0.15$ | $(4.2 \pm 2.9) \times 10^{-2}$ |
| $\beta_{\mathrm{in-cl}}$, $-$ | (14) | SO2SO4EXP | $0.2-3.0$ | $0.90 \pm 0.37$ |
| $k_{\mathrm{SO_2,dry}}$, s$^{-1}$ | (4) | SO2DRYREMCOEF | $(2.0-9.9) \times 10^{-6}$ | $(5.6 \pm 2.3) \times 10^{-6}$ |
| $k_{\mathrm{SO_4,dry}}$, s$^{-1}$ | (4) | SO4DRYREMCOEF | $(0.5-5.0) \times 10^{-7}$ | $(3.6 \pm 1.0) \times 10^{-7}$ |
| $k_{\mathrm{SO_4,wet},0}$, s$^{-1}$ | (15) | SO4WETREMCOEF | $(2.0-9.0) \times 10^{-6}$ | $(6.7 \pm 1.5) \times 10^{-6}$ |
| $p_0$, cm day$^{-1}$ | (15) | PRSO4 | $2-12$ | $4.8 \pm 1.8$ |
| $H_{\mathrm{SO_2}}$, km | (2) | VSCALESO2 | — | 1.2 |
| $H_{\mathrm{SO_4}}$, km | (2) | VSCALEAERO | — | 1.8 |
| $E_{\mathrm{SO_2,min}}$, kgS m$^{-2}$ s$^{-1}$ | — | ESO2MIN | — | $1 \times 10^{-21}$ |
| $n_{\mathrm{smo}}$ | — | NSMOCHAP | — | 5 |





**Table 2.** Global sulphur budget in year 1990 as simulated by ChAP in comparison to other available estimates. The ACCMIP estimates are in square brackets and are either from (Myhre et al., 2013, their Table 4) (sulphate burden) or from (Lamarque et al., 2013a) (emission and depositions). The values in round brackets are from the CMIP5 database (https://tntcat.iiasa.ac.at/RcpDb/). Estimates in quotes are from Table 5.5 of IPCC TAR (Houghton et al., 2001) ascribed to year 1990. Single quotes show the values as they are reported in this Table, and double quotes are for the quantities which are rescaled by the ratio of our emissions in 1990 and the mean IPCC TAR (0.65=63.8/98.2).

| variable | value | | |
|---|---|---|---|
| | 1980 | 1990 | 2000 |
| $E_{\mathrm{SO_2}}$, $\mathrm{TgS\,yr^{-1}}$ | 65.1 | 63.8 | 53.7 |
| | [63.0] | | [53.3] |
| | | '82.5-125.6' | |
| $R_{\mathrm{SO_4,prod}}$, $\mathrm{TgS\,yr^{-1}}$ | 31.3 | 30.7 | 25.6 |
| | | "25.2-47.9" | |
| $D_{\mathrm{SO_2,dry}}$, $\mathrm{TgS\,yr^{-1}}$ | 33.8 | 33.2 | 28.1 |
| | | "10.5-39.1" | |
| $D_{\mathrm{SO_4,dry}}$, $\mathrm{TgS\,yr^{-1}}$ | 4.6 | 4.5 | 3.6 |
| | | "2.6-11.0" | |
| $D_{\mathrm{SO_4,wet}}$, $\mathrm{TgS\,yr^{-1}}$ | 26.7 | 26.1 | 22.0 |
| | [26.7] | | [24.4 ] |
| | | "22.0-37.4" | |
| $D_{\mathrm{SO_x,dry}}$, $\mathrm{TgS\,yr^{-1}}$ | 38.4 | 37.7 | 31.7 |
| | [35.7] | | [25.6] |
| | | "20.2-50.1" | |
| $D_{\mathrm{SO_x,wet}}$, $\mathrm{TgS\,yr^{-1}}$ | 26.7 | 26.1 | 22.0 |
| | [28.7] | | [27.0] |
| | | "22.0-37.4" | |
| $B_{\mathrm{SO_2}}$, $\mathrm{TgS}$ | 0.19 | 0.19 | 0.16 |
| | | "0.13-0.41" | |
| $B_{\mathrm{SO_4}}$, $\mathrm{TgS}$ | 0.41 | 0.40 | 0.32 |
| | (0.41) | (0.39) | (0.38) |
| | | | [0.3–0.9] |
| | | "0.36-0.71" | |
| $\mathcal{T}_{\mathrm{SO_2}}$, days | 1.1 | 1.1 | 1.1 |
| | | '0.6–2.6' | |
| $\mathcal{T}_{\mathrm{SO_4}}$, days | 4.8 | 4.8 | 4.5 |
| | | '3.6–7.2' | |





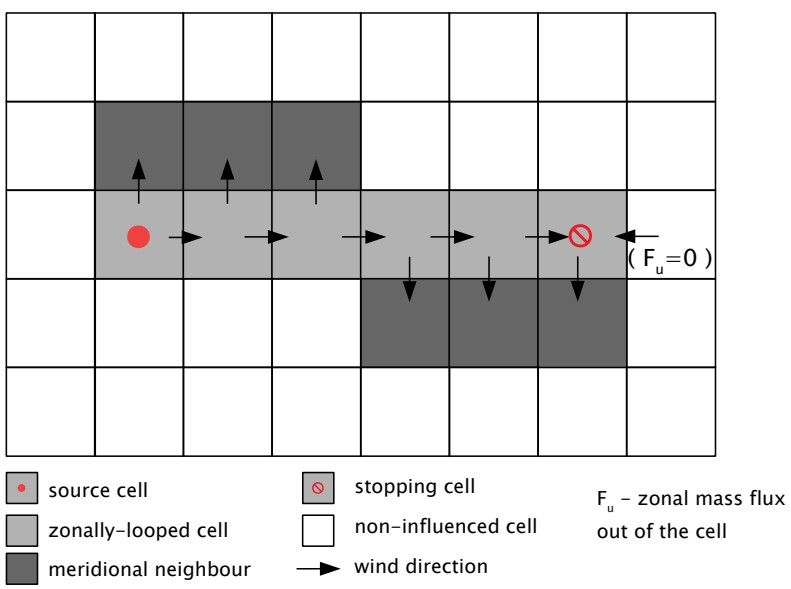

**Figure 1.** A cartoon to illustrate advection in ChAP. Only the case when the stopping grid cell corresponds to the change of the zonal velocity sign is shown.

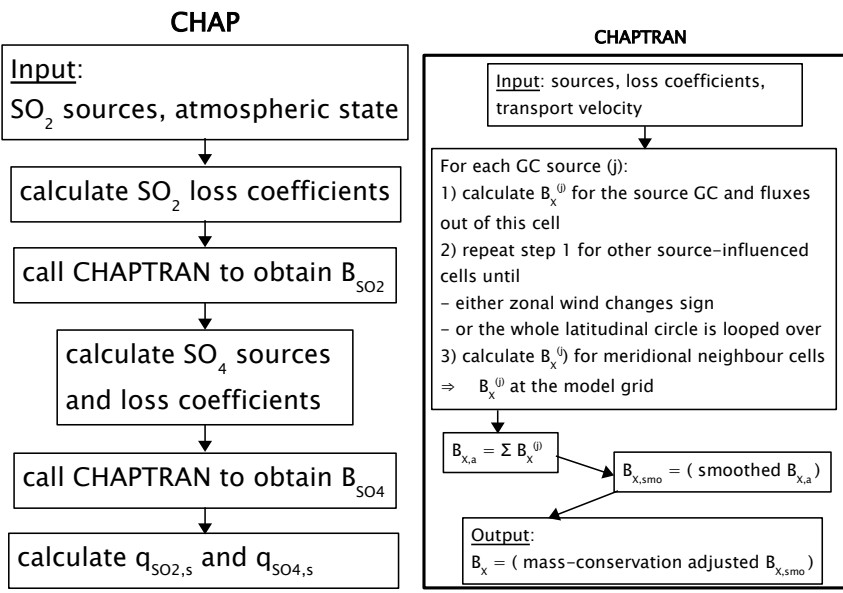

**Figure 2.** A cartoon for ChAP data flow. GC is an acronym for grid cell.



**Figure 3.** The globally and annually averaged modelled $SO_2$ mass in the atmosphere (a), and the respective annual mean (b, c), December-February mean (d, e), June-August mean (f, g) burdens per unit area in 1990 (b, d, f) and 2000 (c, e, g). Green box shows the respective emission-rescaled IPCC TAR Table 5.5 estimate (see text) and its median.



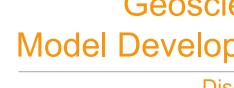
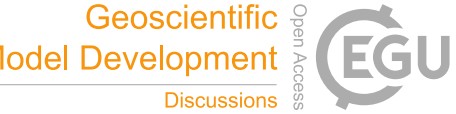

**Figure 3.** (continued)







**Figure 4.** The modelled near–surface concentration of sulphur dioxide in 1990 (a, c, e) and 2000 (b, d, f). Top, middle, and bottom rows show, correspondingly, annual means, and the averages for boreal winter and for boreal summer.





**Figure 5.** The globally and annually averaged modelled $SO_4$ mass in the atmosphere (a), and annual mean burdens per unit area (b-e) in the model (b, d) and in the CMIP5 database (c, e) in years 1990 (b, c) and 2000 (d, e). In panel a, horizontal lines on the colour boxes depict corresponding medians. The ACCMIP data are taken from (Myhre et al., 2013, their Table 4). The IPCC TAR data are adopted from their Table 5.5 and are emission-rescaled (see text).

**b) model** $B_{\mathrm{SO_4}}$**, DJF 1990**

**c) CMIP5** $B_{\mathrm{SO_4}}$**, DJF 1990**

**b) model** $B_{\mathrm{SO_4}}$**, DJF 2000**

**c) CMIP5** $B_{\mathrm{SO_4}}$**, DJF 2000**

**Figure 6.** December-February mean sulphate burdens per unit area in the model (a, c) and in the CMIP5 database (b, d) in years 1990 (a, b) and 2000 (c, d).



**b) model $B_{\mathrm{SO_4}}$, JJA 1990**

**c) CMIP5 $B_{\mathrm{SO_4}}$, JJA 1990**

**b) model $B_{\mathrm{SO_4}}$, JJA 2000**

**c) CMIP5 $B_{\mathrm{SO_4}}$, JJA 2000**

$\mathrm{mgS\,m^{-2}}$

0.5    1    2    5    10

**Figure 7.** Similar to Fig. 6, but for means over June-August.

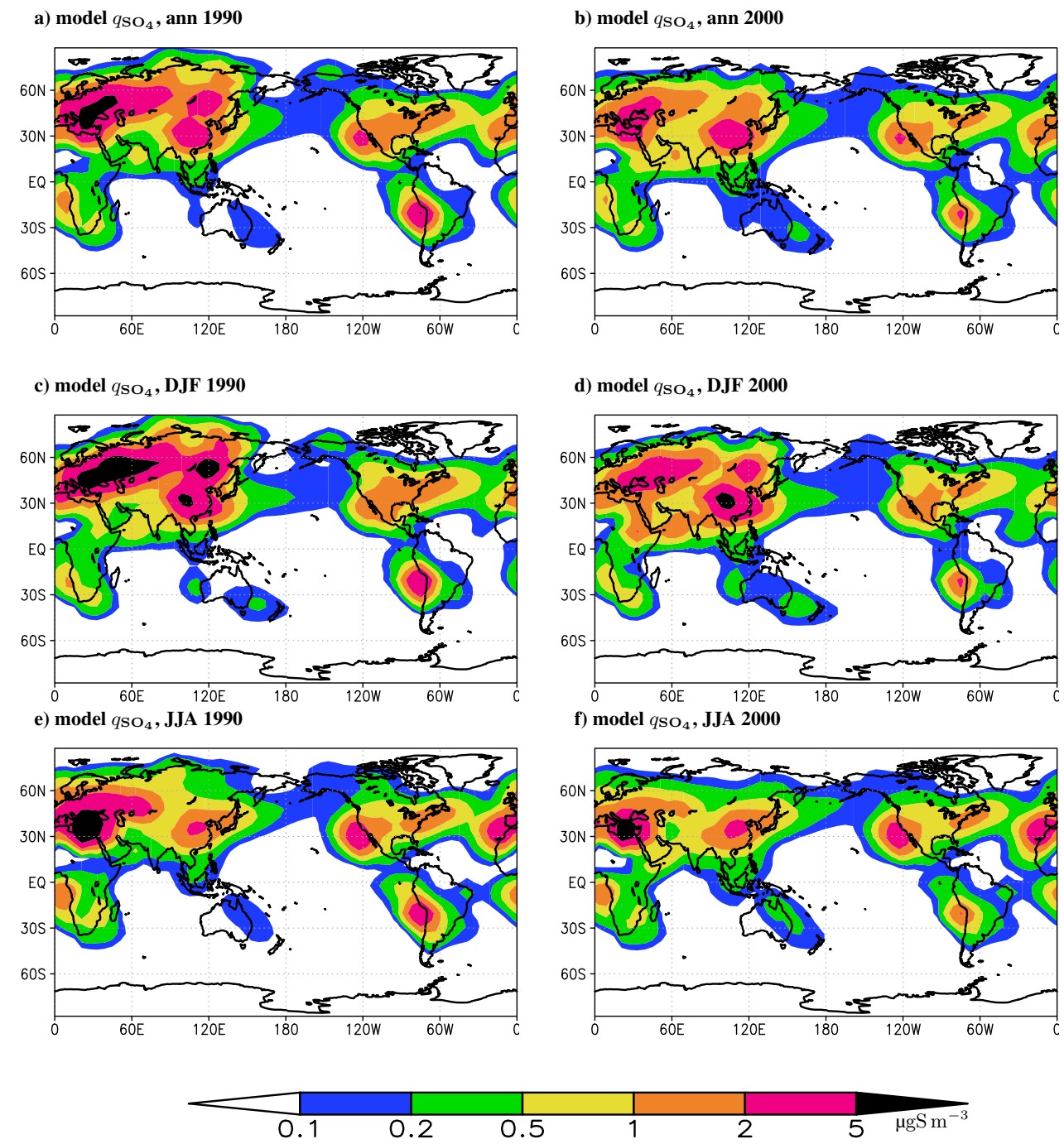

**Figure 8.** Similar to Fig. 4, but for the near–surface $q_{SO_4}$ concentration.





**a) global SO$_x$ deposition**

**b) model $D_{SO_x}$, ann 2000**

**c) ACCMIP $D_{SO_x}$, ann 2000**

**Figure 9.** Global SO$_x$ deposition (a) as well as total (b, c), wet (d, e), and dry (f, g) SO$_x$ depositions per unit area in the model (b, d, f) and in the ACCMIP phase II simulations (c, e, g) for year 2000. In panel a, horizontal lines on the colour boxes depict corresponding medians. The ACCMIP data are taken from (Lamarque et al., 2013a). The IPCC TAR data are adopted from their Table 5.5 and are emission-rescaled (see text); their dry and wet contributions are plotted at different 5-year intervals near year 1990 for visual purposes.

**d) model $D_{\mathbf{SO_x},\mathrm{wet}}$, ann 2000**

**e) ACCMIP $D_{\mathbf{SO_x},\mathrm{wet}}$, ann 2000**

**f) model $D_{\mathbf{SO_x},\mathrm{dry}}$, ann 2000**

**g) ACCMIP $D_{\mathbf{SO_x},\mathrm{dry}}$, ann 2000**

$$0.1 \quad 0.2 \quad 0.5 \quad 1 \quad 2 \qquad \mathrm{MgS\,m^{-2}\,yr^{-1}}$$

**Figure 9.** (continued)

.