# Peer review of "ChAP 1.0: A stationary tropospheric sulphur cycle for Earth system models of intermediate complexity"

_Geoscientific Model Development, 2021_

## Referee Comment (RC1)

**General comments**

**Scientific significance**

The manuscript represents a substantial contribution to modelling science within the scope of this journal. A method is proposed to combine the quality of the calculation and the minimum time required to perform the calculations.

The manuscript presents a substantiated stationary computational scheme for CHAP-1.0 (Chemical and aerosol processes, version 1.0) for modeling the sulfur cycle in the troposphere of the earth. The scheme is designed for models of medium complexity (EMIC) and takes into account the emission of sulfur dioxide to the atmosphere, its deposition on the surface, oxidation to sulfates, as well as dry and wet deposition of sulfates on the surface.

ChAP-1.0 implements only the anthropogenic part of the atmospheric sulphur cycle, but authors plan to extend the scheme in future.

**Specific comments**

**Scientific quality**

The calculations with the scheme are performed forced by anthropogenic emissions of sulphur dioxide into the atmosphere for 1850-2000 adopted from the CMIP5 dataset and by the ERA-Interim meteorology assuming that natural sources of sulphur into the atmosphere remain unchanged during this period. The ChAP output is compared to changes of the tropospheric sulphur cycle simulations: with the CMIP5 data, with the IPCC TAR ensemble, and with the ACCMIP phase II simulations.

In addition, in regions of strong anthropogenic sulphur pollution, ChAP results are compared to other data, such as the CAMS reanalysis, EMEP MSC-W, and with individual model simulations.

However, as can be seen from the comparison of the modeling results with the data of CMIP5, IPCC TAR and with II ACCMIP, (Fig. 5,7, 9), there is a systematic excess of the model values of concentrations by several times in the territories in the western and central parts of Eurasia, South America. Is this related to the peculiarities of the advection calculation scheme, since it is possible to imagine a situation in which the flow within the cell moves in opposite directions and the average velocity over the cell will be zero?

**Scientific reproducibility**

The description given in the manuscript as a whole allows other scientists to reproduce the simulation.

**Technical corrections**

**Presentation quality**

Numbering in Fig. 5-7 do not correspond to the figure captions - there are no designations d, e.

---

## Referee Comment (RC2)

Review of GMD-2021-24 - ChAP 1.0: A stationary tropospheric sulphur cycle for Earth system models of intermediate complexity.

This paper describes a simplified model of the Sulphur cycle that can be implemented in Earth system models of intermediate complexity. The scheme appears to be very fast and to be able to reproduce the broad patterns of $SO_2$ and $SO_4$ burden and surface concentration. The paper is well written, and describes in details all of the aspects of the scheme, as well as the tuning method. The fact that ChAP 1.0 is very fast to run allows for an easy tuning of all parameters, and make it well suited also for sensitivity studies. A number of hypothesis and simplifications are made; most of them are discussed in Section 6. I think the paper reaches its objectives in describing ChAP 1.0 as well as its limitations, and providing a basic validation of its approach compared to more complex models. I have a few minor remarks, as well as a few suggestions (which can tried later on and don't have to be mentioned in the manuscript, if the authors think them useful). Remarks:

- I may have missed it but I didn't find the information about what time step was used in the simulations shown in the manuscript: if it is not present, could you please add it?

- In the discussion of the limitations (Section 6), I think two major assumptions should be at least mentioned: that of the fixed lifetime of $SO_2$/$SO_4$ as well as the vertical length scale. While the values chosen appear sensible and in line with results from more complex models, the fact that the spatio-temporal variations of these parameters is not accounted for could have an impact on the results of ChAP1.0. The vertical length scale for example probably varies a lot between day and night (in clear-sky conditions), while the lifetime of $SO_4$ is heavily impacted by its main sink, wet deposition, and in turn by the occurrence of precipitation. It is possible that the tuning stage compensated partly for not taking these into account. (and the other hypothesis outlined in Section 6).

- The tuning procedure (Section 4, line 240): where does the observed $SO_4$ burden per unit come from?

Suggestions:
- For $SO_2$ lifetime, the authors may think of using the very simple parameterization from Huneeus et al. (2007), as a function of latitude only: (from Remy et al. : 2019): "The conversion rate (per second) can be written as C0 = exp   − δt (C1−C2 cos θ )   δt , (16) where δt is the time step, θ is the angular latitude, and C1 and C2 are e-folding times in days representing the lifetime at the pole and the Equator set to 8 and 5 days, respectively, for operational cycles up to 43R1."

- For dry deposition: to use different values over ocean and land (and possibly, ice/snow). That would be quite simple to implement and test and could give a bit more variability to the model.

- For wet deposition, to distinguish between solid and liquid precipitations, ie to split $k_{SO4,wet}$ in $k_{SO4,wetrain}$ and $k_{SO4,wetsnow}$, and then compute $k_{SO4,wetrain,0}$ and $p_0$ specifically for both rain and snow. Wet deposition by snow is generally much less intense than by rain, so this again could make a difference.

---

## Author Comment (AC1)

**Reply to the reviewer's comments to**
**ChAP 1.0: A stationary tropospheric sulphur cycle for Earth system models of intermediate complexity**

A. V. Eliseev, R.D. Gizatullin, and A.V. Timazhev

October 26, 2021

We are grateful for the reviewer for the constructive comments which led to the improved presentation of our results.

The most important changes in the manuscript are as follows:

- The second referee's suggestions are discussed as limitations of the contemporary version of the scheme and noted as routes to improve it.

- In addition, an issue of distinguishing heavy and light rain in wet deposition calculations is noted and discussed.

- Some typos (in particular those found by the reviewer) are corrected.

This reviewer made only one editorial comment, *"Numbering in Figs. 5-7 do not correspond to the figure captions – there are no designations d, e"* .
This was a misprint: items 'b' and 'c' in panel titles were erroneously duplicated. Upon revision, this misprint is corrected.

---

## Author Comment (AC2)

**Reply to the reviewer's comments to* ChAP 1.0: A stationary tropospheric sulphur cycle for Earth system models of intermediate complexity**

A. V. Eliseev, R.D. Gizatullin, and A.V. Timazhev

October 26, 2021

We are grateful for the reviewer for the constructive comments which led to the much improved presentation of our results. In addition, we are grateful to suggestions for future work – all they open important routes for improving our scheme.

The most important changes in the manuscript are as follows:

- Referee's suggestions are discussed as limitations of the contemporary version of the scheme and noted as routes to improve it.

- In addition, an issue of distinguishing heavy and light rain in wet deposition calculations is noted and discussed.

- Some typos (in particular those found by the first reviewer) are corrected.

Below, the point-to-point replies to the comments and suggestions are listed. Original comments and suggestions are typed in italic.

**Comments**

- *I may have missed it but I didn't find the information about what time step was used in the simulations shown in the manuscript: if it is not present, could you please add it?*
  The stationary approximation embedded into ChAP removes the necessity to specify the time step, and time stepping is completely determined by the monthly mean forcing data. The corresponding note is added to Sect. 3 of the manuscript.

- *In the discussion of the limitations (Section 6), I think two major assumptions should be at least mentioned: that of the fixed lifetime of $SO_2/SO_4$ as well as the vertical length scale. While the values chosen appear sensible and in line with results from more complex models, the fact that the*

*spatio-temporal variations of these parameters is not accounted for could
have an impact on the results of ChAP1.0. The vertical length scale for
example probably varies a lot between day and night (in clear-sky condi-
tions), while the lifetime of $SO_4$ is heavily impacted by its main sink, wet
deposition, and in turn by the occurrence of precipitation. It is possible
that the tuning stage compensated partly for not taking these into account
(and the other hypothesis outlined in Section 6).*

Apart from the apparently simplistic formulations of the conversion and
deposition rates (see Sect. "Suggestions"), the first assumption was not
used in the tuning procedure. Contrary to the previously available scheme
for the tropospheric sulphur cycle designed for EMICs (Bauer et al., 2008),
our scheme does not employ an assumption of fixed lifetimes for both $SO_2$
and $SO_4$. In ChAP, both lifetimes are determined by the conversion and
deposition coefficients which depend on climate and on burden of the
compounds coming from the earlier steps of chemical chains. We note
to non-systematic variations of both lifetimes between different simulated
time slices. The respective statement is added to Conclusions.

However, our scheme does employ an explicit assumption of fixed vertical
scales for $SO_2$ and $SO_4$. We agree that it should be properly discussed as
a limitation of the contemporary ChAP code. The respective discussion in
Sect. 6 is extended upon revision. In addition, this limitation is explicitly
listed in Conclusions, where it was missed in the previous version of the
manuscript.

- *The tuning procedure (Section 4, line 240): where does the observed $SO_4$
  burden per unit come from?*
  Yes, this issue was missed in the previous version of the paper. In the
  revised version, it is stated explicitly in Sect. 4 that we used the CMIP5
  sulphate burdens per unit area in place of $B_{SO_4,o}$ in Eq. (16).

**Suggestions**

- *For $SO_2$ lifetime, the authors may think of using the very simple param-
  eterization from Huneeus et al. (2007), as a function of latitude only:
  (from Remy et al. : 2019): "The conversion rate (per second) can be
  written as*

$$C_0 = \frac{\exp\left[-\frac{\delta t}{(C_1 - C_2 \cos\theta)}\right]}{\delta t},$$

  *(16), where $\delta t$ is the time step, $\theta$ is the angular latitude, and $C_1$ and
  $C_2$ are e-folding times in days representing the lifetime at the pole and
  the Equator set to 8 and 5 days, respectively, for operational cycles up to
  43R1."*
  Yes, it would de a valuable option to parametrise an impact of the OH
  abundance on oxidation rate of $SO_2$. The respective note and a corre-
  sponding reference is added to Sect. 6 of the manuscript.

- *For dry deposition: to use different values over ocean and land (and possibly, ice/snow). That would be quite simple to implement and test and could give a bit more variability to the model.*

  We agree that it is sensible to prescribe dry deposition rate coefficients as a function of land surface type with a distinction between the open ocean, snow/ice, and land without ice and snow. This possibility is omitted on purpose in the present manuscript. The reasoning behind this choice is due to i) the neglect of the oceanic sources of sulphur which directly hampers tuning of $k_{SO_2,dry}$ and $k_{SO_4,dry}$ over the ocean, and ii) an attempt to demonstrate the ability of the present, simplistic version of ChAP to reproduce large-scale properties of the sulphur compounds distribution in the atmosphere. Nonetheless, we opt to try this option in future. The corresponding discussion is added to Sect. 6.

- *For wet deposition, to distinguish between solid and liquid precipitations, i.e. to split $k_{SO_4,wet}$ in $k_{SO_4,wetrain}$ and $k_{SO_4,snow}$, and then compute $k_{SO4,wetrain,0}$ and $p_0$ specifically for both rain and snow. Wet deposition by snow is generally much less intense than by rain, so this again could make a difference.*

  We agree that contemporary implemented formulation (Eq. (15)) does not distinguish between different precipitation types: light rain, heavy rain, and snow. Light and heavy rains show principally different efficiencies for removing hygroscopic aerosols from the atmosphere. Snow is an inefficient aerosol remover as well. The work to implement a distinction between different precipitation types in our scheme is under way and is expected to be implemented into the next version of ChAP.